# Assessing a chip based rapid RTPCR test for SARS CoV-2 detection (TrueNat assay): A diagnostic accuracy study

**Ujjala Ghoshal, Atul Garg**  *, **Shruthi Vasanth, Akshay K. Arya, Ankita Pandey, Nidhi Tejan, Vikas Patel, Vikram P. Singh**

Department of Microbiology, Sanjay Gandhi Post Graduate Institute of Medical Sciences, Lucknow, India

* atulgargsgpgi@gmail.com

## Abstract

COVID-19 testing is required before admission of a patient in the hospitals, invasive procedures, major and minor surgeries etc. Real Time Polymerase chain reaction is the gold standard test for the diagnosis, but requires well equipped biosafety laboratory along with trained manpower. In this study we have evaluated the diagnostic accuracy of novel True-Nat molecular assay for detecting SARS CoV-2. TrueNat is a chip-based real time PCR test and works on portable, light weight, battery powered equipment and can be used in remote areas with poor infrastructure. In this study 1807 patients samples were collected for both TrueNat and RTPCR COVID-19 testing during study period. Of these 174 (9.7%) and 174 (15%) were positive by RTPCR and TrueNat respectively and taking results of RTPCR as gold standard TrueNat test showed a sensitivity, specificity and diagnostic accuracy of 69.5, 90.9% and 89.2% respectively. It can be concluded that TrueNat is a simple, easy to use, good rapid molecular diagnostic test for diagnosis of COVID-19 especially in resource limited settings and will prove to be a game changer of molecular diagnostics in future.

## Introduction

The on-going COVID19 pandemic; caused by severe acute respiratory syndrome coronavirus-2 (SARS CoV-2) was declared as pandemic on 11.03.2020 by World Health Organization (WHO) [1]. Since then till 20.02.2020 it has affected globally approximately 110 million cases and 2.5 million deaths.

COVID-19 testing is required before admission of a patient in the hospitals, invasive procedures, major and minor surgeries etc. Real Time Polymerase chain reaction (RTPCR) is the gold standard test for the diagnosis, however its turnaround time is 6–8 hours and requires well equipped biosafety laboratory level-II along with trained manpower. In cases where early and rapid diagnosis of COVID-19 is warranted RTPCR testing leads to delay in diagnosis along with tension and anxiety both among patient and treating health care workers. So a rapid, cheap molecular diagnostic test with high sensitivity and specificity is urgently required for detecting SARS CoV-2, especially in developing countries and in rural area where there is poor infrastructure and lack of well-equipped labs [2].

**Data Availability Statement:** All relevant data are within the manuscript and its Supporting Information files.

**Funding:** The authors received no specific funding for this study.

**Competing interests:** The authors have declared that no competing interests exist.

In this study we have evaluated the diagnostic accuracy of TrueNat assay, a chip based rapid molecular diagnostic test for detecting SARS CoV-2. This technology is based on portable, light weight, battery powered, TaqMan probe based Real time polymerase chain reaction technology developed and manufactured by Molbio Diagnostics Private Limited, Goa, India. The TrueNat device can be used for detection of more than 25 pathogens like malaria, tuberculosis, hepatitis B, HIV, dengue, H1N1 influenza, chikungunya, Rabies, Influenza, SARS Cov-2 etc [3]. In 2018 TrueNat technology was adopted by Revised National Tuberculosis Control Programme (RNTCP) for tuberculosis diagnosis in India and in 2020 it was also endorsed by World health organization for diagnosis of Multi drug resistant tuberculosis [4].

This equipment is a laboratory in a suitcase and can be used in remote areas with poor power supply and connectivity. The device has an automated reporting system and is GPRS/Bluetooth enabled, to aid in result data transfer. The TrueNat machine is available in three different models UnoDx, Duo, and Quattro, with capacity to test one, two, and four samples per run, respectively [5]. At our center we have used the Quattro machine, this equipment set cost approximately USD 18,000 and running cost per test is USD 15 only.

In April, 2020 the TrueNat test for diagnosis of SARS CoV-2 was launched by Molbio Diagnostics and was subsequently approved by Indian apex medical research organization—Indian Council of Medical Research (ICMR), New Delhi [6]. The manufacturer claims a sensitivity and specificity of 100% and 98.8% respectively but currently there are no data on diagnostic accuracy in field setting [5]. Thus this study was planned with aims to evaluate the diagnostic accuracy of TrueNat test for the diagnosis of SARS CoV-2 as compared to RTPCR; the gold standard reference test.

## Material and methods

### Study site and population

This retrospective observational study was designed and conducted at the Department of Microbiology, Sanjay Gandhi Post Graduate Institute of Medical Sciences (SGPGIMS), Lucknow, India. This study was approved by the Institutional Ethics Committee (IEC) and being retrospective in nature the IEC waived the requirement for patient consent by approval number 2020-279-IP-EXP-31. All study data was retrieved from hospital information System of SGPGIMS.

The samples were collected at triage of a dedicated COVID-19 tertiary care center with 180 beds including 30 ICU ventilator beds. For TrueNat testing single oropharyngeal swab was collected in viral lysis media (VLM) provided by Molbio diagnostics Ltd, whereas for RTPCR both oropharyngeal and nasopharyngeal swabs were collected in viral transport media (HI media Labs, India) and transported to the COVID-19 laboratory in a cold chain. At time of COVID-19 pandemic peak (August–October 2020) any patient requiring treatment at SGPGIMS, Lucknow had to undergo mandatory COVID-19 testing and all cases in which both samples for RTPCR and TrueNat were collected within time gap of less than 24 hours by treating physician were included in this study.

### TrueNat workstation

The TrueNat workstation consist of a nucleic acid extraction device (Trueprep AUTO V2) and a real time polymerase chain reaction analyzer (Truelab Uno Dx/Uno/Quattro), along with accessories such as a RNA cartridges, TrueNat Chips, micro tip holding stand etc. Both devices are portable, powered by a rechargeable batteries and can run continuously for $\geq 6$ hours on single charge. Trueprep AUTO V2 is fully automated RNA extractor and uses a disposable fluidic cartridge to extract RNA from VLM within 15 minutes. 6 μL of extracted RNA is added to

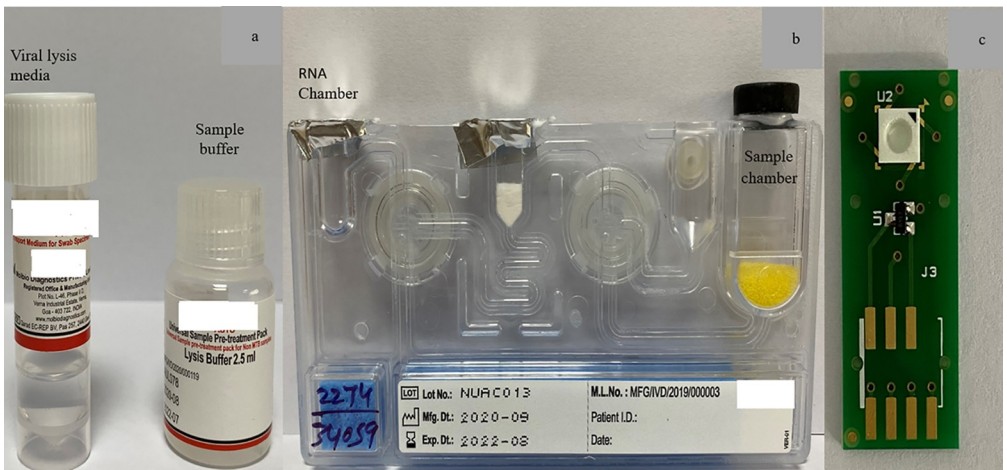

**Fig 1. Heading: Images of consumables used in the TrueNat assay.** Legend: (1a) Viral Lysis media and the sample buffer. (1b) RNA extraction cartridge. (1c) Micro PCR chip.

a PCR tube consisting of room temperature stabilized real time PCR reagents and the mixture is added onto a disposable microchip. The chip is loaded in Truelab analyzer and programme is selected for appropriate assay. COVID -19 testing was done using two step strategy; all samples are initially tested by E gene assay and all positive samples with cut off threshold of <32 are confirmed by RNA dependent RNA polymerase (RdRp) gene assay [6]. All samples that are positive by RdRP assay with Ct value < 32 are considered as true positives. We used Truelab Quattro model in this study. Four samples can be processed at a time taking about 1 hour, including sample preparation. One device can process 64 samples per day in a 24 hour working laboratory.[Figs 1–3]

## COVID-19 detection by real time PCR

**RNA extraction.** RNA extraction was performed on 200 $\mu$l of viral transport media using a QIAamp RNA mini kit (Qiagen, Inc., Valencia, Calif.) as per manufacturer's instructions.

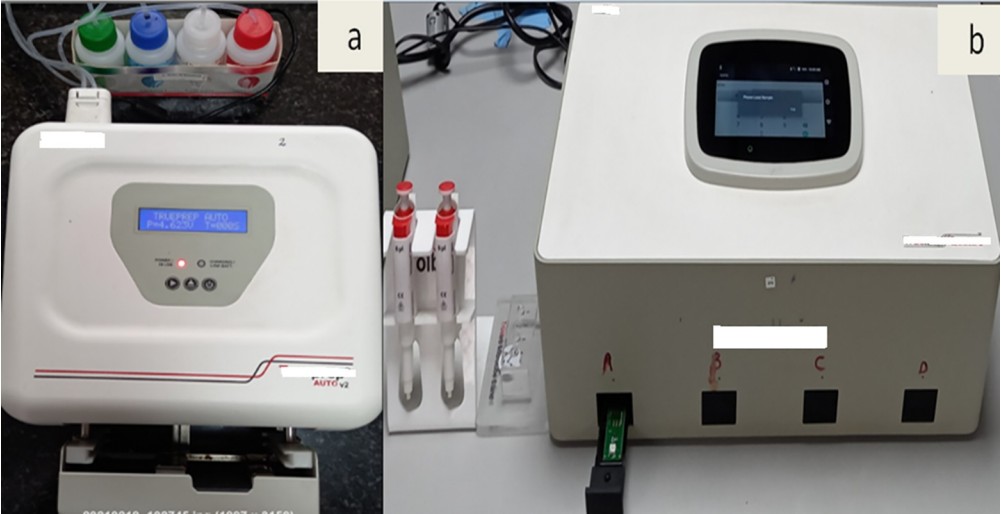

**Fig 2. Heading: Images of TrueNat equipment.** Legend: (2a) Trueprep AUTO v2 Sample Prep Device. (2b) Quattro Real Time micro PCR Truelab Analyzer.

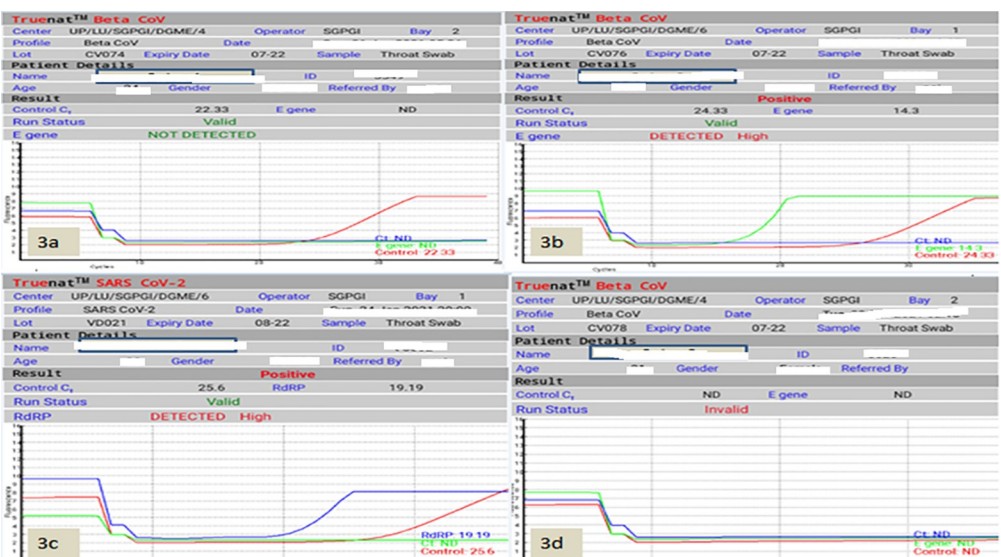

**Fig 3. Heading: Shows the RTPCR graph generated by TrueNat machine.** Legend: 3a shows a negative report with only fluorescence in the Control gene RNase P; 3b shows the fluorescence in the E-gene and control gene; 3c shows fluorescence above the cutoff to RdRP and control gene; 3d shows an invalid test result with no amplification in RNase gene.

**Qualitative real time PCR.** A 25 μL reaction was prepared for detection of SARS CoV-2 by RTPCR utilizing 5 μL of extracted RNA, 12.5 μL of 2X PCR buffer, 1 μL of Primer and Taq-Man probe sequences targeting E genes, RdRP and RnaseP as per W.H.O protocol [7]. The thermal cycling was performed at 55˚C for10 min for reverse transcription, followed by 95˚C for 3 min and then 40 cycles of 95˚C for 15 s, 58˚C for 30s using Quant Studio 5 Real Time PCR system (Thermo Fisher Scientific, Massachusetts USA). All samples were screened for E gene and positive samples were confirmed by detection of specific RdRP gene. Cut off threshold (Ct value) <40 were considered as positive.

**Statistical analysis.** The sensitivity, specificity, positive predictive value, negative predictive value, diagnostic accuracy and 95% Confidence Intervals were calculated SPSS software version 23.0 (SPSS, Inc., Chicago, USA)

## Results and discussion

Among 1807 patients samples for both TrueNat and RTPCR COVID-19 testing were collected during study period. Of these 174 (9.7%) and 174 (15%) were positive by RTPCR and TrueNat respectively. A total of 121 (6.7%) samples were positive by both RTPCR and TrueNat test; 149 (8.2%) samples were TrueNat positive and RTPCR negative, 53 (3.0%) samples were TrueNat negative and RTPCR positive and 1484 (82.1%) were negative. (Table 1) Taking results of RTPCR as gold standard TrueNat test showed a sensitivity, Specificity and diagnostic accuracy of 69.5, 90.9% and 89.2% respectively. (Table 2)

Detailed analysis of 149 samples that were TrueNat positive and RTPCR negative revealed that the mean cut off threshold (Ct value) of E gene among TrueNat positive samples was 27 and in 50 (33%) samples Ct value was high (>30). Similarly detailed analysis of 53 samples that were TrueNat negative and RTPCR positive showed that of these 18 samples were TrueNat positive for E gene with CT value more than 32, but were considered negative as per TrueNat interpretation guidelines. (S1 Data) Overall wastage in TrueNat processing was also calculated and it was found that while processing 1807 samples; 50 RNA cartridges (2.7%) and 18 True-Nat Chips (1%) were wasted.

**Table 1. Comparison of the positivity rates of TrueNat with RTPCR results.**

|  | RTPCR positive | RTPCR negative | Total |
|---|---|---|---|
| **TrueNat Positive** | 121 | 149 | 270 |
| **TrueNat negative** | 53 | 1484 | 1537 |
| **Total** | 174 | 1633 | **1807** |

Detection of SARS CoV-2 by RTPCR is the current gold standard for diagnosis of COVID-19. World health organization has repeatedly stressed the importance of the molecular diagnosis of COVID-19 for prompt management of patients, isolation and contact tracing and limit its spread. Point of care test has been identified by a WHO expert group as the first of eight research priorities in response to the COVID-19 outbreak [8] and play an important role in medical emergencies like myocardial infarction, acute abdomen, emergency surgeries and other medical emergencies where urgent intervention is required.

Accurate and timely results are back bone of decision making, both in the inpatient and OPD settings. Quick turnaround time of test reports is also critical for prudent use of resources, such as the availability of emergency and Triage area beds, isolation rooms and real-time cohorting decisions. However for performing RTPCR testing a fully functional air-conditioned biosafety level-2 microbiology laboratory is required; equipped with specialized instruments like biosafety cabinets class II, automated RNA extractors, Real time PCR machine and trained manpower to process the samples while ensuring biosafety and bio security [9]. It is challenging to set up a fully functional molecular testing laboratory in resource limited developing countries where availability of high end equipments, trained manpower, uninterrupted power supply are a big problem and at such places newer options should be explored.

There is an urgent need for a nucleic acid-based COVID-19 test that is highly sensitive and specific, and can be used at point-of-care in resource-limited settings. In April 2020 Molbio Diagnostics Ltd, Goa, India introduced TrueNat COVID-19 testing; It is portable, battery-operated point of care molecular diagnostic test especially designed for areas with low resources. The device has an automated reporting system and is GPRS/Bluetooth enabled, to aid in result data transfer. The TrueNat manufacturer claims a sensitivity and specificity of 100% with lower limit of detection at 407 copies. Till date there are no field trials of SARS CoV-2 molecular detection by TrueNat; thus this study was planned to evaluate the diagnostic accuracy of TrueNat test for the diagnosis of SARS CoV-2 as compared to gold standard test RT-PCR.

The results of this study showed a sensitivity of 69.5%, Specificity of 90.9%, NPV- 97.2% and diagnostic accuracy of 89.2%. In another recent laboratory based study performed on pre characterized archived samples study by ICMR, New Delhi; 75 samples (30 positives and 45 negatives) were tested with TrueNat for SARS Cov-2 and were found to be 100% sensitive and specific with 100 copies as lower limit of detection [10]. However in this study the researchers used viral transport media consisting of both nasopharyngeal and oropharyngeal swabs instead of recommended viral lysis media containing only oral swab. Further only 30 positive samples

**Table 2. Showing statistical analysis of study data.**

| S No | Statistic | Value | 95% CI |
|---|---|---|---|
| 1 | Sensitivity | 69.5% | 59.2% - 76.5% |
| 2 | Specificity | 90.88% | 89.3% to 92.3% |
| 3 | Positive Predictive Value (PPV) | 39.8% | 35.6% to 44.3% |
| 4 | Negative Predictive Value (NPV) | 97.2% | 96.5% to 97.7% |
| 5 | Diagnostic Accuracy | 89.2% | 64% to 90.5% |

were tested and these reasons might have contributed to high sensitivity and specificity of 100%. TrueNat has also been evaluated for other infectious diseases. In a study on human papilloma virus detection in cervical samples TrueNat showed sensitivity and specificity of 97.7% and 98.9%, respectively compared to conventional test. In another study targeting malaria parasite TrueNat was 99% sensitive compared to gold standard microscopy. Recently it has been approved by WHO for testing drug resistant tuberculosis [11–14].

As per Ministry of Health and family welfare, India; currently TrueNat COVID-19 testing is performed at 886 centers across India, 632 in government sector and 254 in private sector. Of these more than 50% machines are installed in remote distant places with poor infrastructure; TrueNat COVID-19 testing is available at remote places like karavati Island, Lakshadweep (500 Kms away from Mainland) and 23 machines are installed in distant north east Indian state of Arunachal Pradesh having a rough hilly terrain [14]. The extensive infrastructure of TrueNat testing has played a key role in control of community transmission of SARS CoV-2. The operational cost of TrueNat (USD 15/Sample) is less compared to Real time PCR and it can provide report in 1 hour compared to 10–12 hour time required in RTPCR. Further the sample is collected in virus lysis media which immediately lyses the infective virus and makes the sample non infectious and removing the requirement of biosafety cabinet and makes it a excellent point of care test having minimal facilities.

There are few limitations of this study. Ideally for evaluating a new diagnostic test it should be performed on same clinical specimens that is tested by reference method; however in this study as per manufacturer's instructions and ICMR, New Delhi, India guidelines TrueNat testing was performed on virus lysis media containing only Oropharyngeal swab while RTPCR was performed on viral transport media containing both oro pharyngeal and nasopharyngeal swabs. In most of the cases both samples were collected simultaneously however in some cases the time gap of two sample collection was 12–24 hour.

Laboratory testing with real time PCR has the advantage of high throughput processing that cannot be achieved by the TrueNat platform. In RTPCR 96 samples can be processed simultaneously; whereas in TrueNat each RNA extraction unit can process only one cartridge at a time, and maximum 4 samples can be tested simultaneously. However, judicious use of TrueNat testing could relieve the burden on central molecular laboratories and increase overall testing capacity, complementing existing approaches. The TrueNat plays a strategic role in places where results can affect real time decision making such as screening trauma victims requiring emergency surgeries, triaging admissions, maternity labor rooms and screening elective admissions or staff (eg, before dialysis or chemotherapy) etc.

## Conclusion

Based on study results it can be concluded that TrueNat is a simple, easy to use, good rapid molecular diagnostic test for diagnosis of COVID-19 and will prove a game changer in molecular diagnostics of infectious disease in future especially in areas with poor Infrastructure.

## Supporting information

**S1 Data.**
(XLSX)

## Acknowledgments

We would like to acknowledge Department of Medical Education, Uttar Pradesh, India for providing the kits and consumables used in the study. We also acknowledge the help of

technical staff Mr. V K Mishra, Mr. Hemant Verma and Mr. Dinesh Gangwar for their help in molecular assays. We also thank Mr. Nikhil Sharma for his data entry and handling.

## Author Contributions

**Conceptualization:** Ujjala Ghoshal.

**Data curation:** Nidhi Tejan.

**Formal analysis:** Shruthi Vasanth.

**Investigation:** Nidhi Tejan, Vikas Patel.

**Methodology:** Akshay K. Arya, Ankita Pandey, Vikram P. Singh.

**Software:** Akshay K. Arya.

**Supervision:** Atul Garg.

**Writing – original draft:** Atul Garg.

**Writing – review & editing:** Ujjala Ghoshal.

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
