## [Decision Letter · Decision Letter 0]

27 May 2021

PONE-D-21-10427

Assessing a chip based rapid  RTPCR test for SARS C0V-2 detection (TrueNat assay): a diagnostic accuracy study.

PLOS ONE

Dear Dr. Garg,

Thank you for submitting your manuscript to PLOS ONE. After careful consideration, we feel that it has merit but does not fully meet PLOS ONE’s publication criteria as it currently stands. Therefore, we invite you to submit a revised version of the manuscript that addresses the points raised during the review process.

We look forward to receiving your revised manuscript.

Kind regards,

Seyed Ehtesham Hasnain, Ph.D

Academic Editor

PLOS ONE

Journal Requirements:

3. We note that Figure 1 in your submission contains copyrighted images. All PLOS content is published under the Creative Commons Attribution License (CC BY 4.0), which means that the manuscript, images, and Supporting Information files will be freely available online, and any third party is permitted to access, download, copy, distribute, and use these materials in any way, even commercially, with proper attribution. For more information, see our copyright guidelines: http://journals.plos.org/plosone/s/licenses-and-copyright.

We require you to either (a) present written permission from the copyright holder to publish this figure specifically under the CC BY 4.0 license, or (b) remove the figure from your submission:

b.  If you are unable to obtain permission from the original copyright holder to publish this figure under the CC BY 4.0 license or if the copyright holder’s requirements are incompatible with the CC BY 4.0 license, please either i) remove the figure or ii) supply a replacement figure that complies with the CC BY 4.0 license. Please check copyright information on all replacement figures and update the figure caption with source information. If applicable, please specify in the figure caption text when a figure is similar but not identical to the original image and is therefore for illustrative purposes only.

Additional Editor Comments:

Major Revision

Reviewers' comments:

Reviewer's Responses to Questions

**Comments to the Author**

1. Is the manuscript technically sound, and do the data support the conclusions?

Reviewer #1: Yes

Reviewer #2: Yes

2. Has the statistical analysis been performed appropriately and rigorously? 

Reviewer #1: Yes

Reviewer #2: Yes

3. Have the authors made all data underlying the findings in their manuscript fully available?

Reviewer #1: Yes

Reviewer #2: Yes

4. Is the manuscript presented in an intelligible fashion and written in standard English?

Reviewer #1: No

Reviewer #2: Yes

5. Review Comments to the Author

Reviewer #1: Comments:

Present study of Ghoshal U et al. is based on diagnostic approach to encounter present pandemic globally. It is a comparative study between to molecular diagnostics and more informative while testing COVID 19 virus. However, manuscript is written in a very casual approach.

Abstract:

• Abbreviation of PCR should write at proper sections (Line no. 30, 31, 33, 35).

• Author can modify the Conclusion section.

Introduction:

• Advised to write the current global as we as India epidemiological data (Line no. 46, 47).

• It is advised to authors to modify the sentence, as it seems duplicated from abstract section (COVID-19 testing is required before admission of a patient…..).

• Advised to write the updated name of RNTCP.

Material and Methods:

• Author should write the abbreviation RTPCR trough out the manuscript (WHO, OPD etc).

• It is advised to write abbreviation in a single way (line no. 46, 47, 129, 130).

• Write NPV and PPV in statistical analysis section also.

Results and Discussion:

• It is advised to author that please recheck the percentage as well as sample number before writing the result.

• Don’t use full stop sign (.) within a single sentence (Line no. 166-169).

• It is advised to avoid the duplication of sentences (Line no. 181- 190).

• Modify the sentence (Further only 30 positive samples were tested and these reasons might have…..).

• Reference should write as per the journal guideline.

• Reference required for actual Operational cost for TrueNat test and RTPCR.

Reviewer #2: This study is in time and address issue of quick and handy detection of the samples which can be helpful in accessing the infection rate and frequency. However i have one major concern which need clarification from author .

Whether author compared the fidelity of this assay with the RAT ( rapid antigen test ) , how correlative and reliable is this assay. Whether the author used simple PCR or modified nested one. Can we identify mutated strain also in the patients.

6. PLOS authors have the option to publish the peer review history of their article (what does this mean?). If published, this will include your full peer review and any attached files.

Reviewer #1: No

Reviewer #2: No

---

## [Author Response · Author response to Decision Letter 0]

19 Jul 2021

POINT WISE ANSWERS TO REVIEWER COMMENTS

Reviewer 1: 

1. Abbreviation of PCR should BE written at proper sections (Line no. 30, 31, 33, 35).

Answer: We have corrected the manuscript as suggested

2. Author can modify the Conclusion section of abstract.

Answer: We have modified the conclusion in abstract.

3. Advised to write the current global as well as India epidemiological data (Line no. 46, 47).

Answer: We have updated the global data and added Indian data as per your suggestion.

4. It is advised to authors to modify the sentence, as it seems duplicated from abstract section (COVID-19 testing is required before admission of a patient…..).

Answer: We have modified the sentence in revised manuscript. 

5. Advised to write the updated name of RNTCP.

Ans: Name updated as National Tuberculosis Elimination Programme.

6. Author should write the abbreviation RTPCR trough out the manuscript (WHO, OPD etc).

Answer: We have incorporated this change in revised manuscript.

7. It is advised to write abbreviation in a single way (line no. 46, 47, 129, 130).

Answer: We have taken care to write abbreviation in single way in revised manuscript.

8. Write NPV and PPV in statistical analysis section also.

Answer: We have incorporated this change in revised manuscript.

9. It is advised to author that please recheck the percentage as well as sample number before writing the result.

Answer: Thank you for pointing out the typing mistake. We have corrected the data in revised manuscript.

10. Don’t use full stop sign (.) within a single sentence (Line no. 166-169).

Answer: We have incorporated this change in revised manuscript.

11. It is advised to avoid the duplication of sentences (Line no. 181- 190).

Answer: While revising the manuscript we have removed the duplicate sentences.

12. Modify the sentence (Further only 30 positive samples were tested and these reasons might have…..).

Answer: We have revised the sentence in revised manuscript.

13. Reference should be written as per the journal guideline.

Answer: Reference formatted as per Journal guidelines.

14. Reference required for actual Operational cost for TrueNat test and RTPCR.

Answer: Reference added in revised manuscript.

Reviewer 2: 

This study is in time and address issue of quick and handy detection of the samples which can be helpful in accessing the infection rate and frequency. However, one major concern which need clarification from author.

1. Whether author compared the fidelity of this assay with the RAT (rapid antigen test), how correlative and reliable is this assay. 

Answer: In this study in small subset of patients (n=30) all three test i.e. antigen detection by STANDARD Q COVID-19 Ag Test (SD Biosensor, South Korea), Truenat test and RTPCR was done. Taking RTPCR results as gold standard; Antigen detection and Truenat test showed sensitivity of 55% and 70 % respectively. The specificity in each test was 100%. Antigen detection missed 2 RTPCR positive cases (Ct value 29 and 31) but were detected by Truenat test. However, as the sample size is small this data is not included in manuscript.

2. Whether the author used simple PCR or modified nested one. 

Answer: We used simple RTPCR protocol suggested by Indian council of medical research, India. All samples were tested for E gene and RdRP gene and cut off threshold <40 was considered as positive. 

3. Can we identify mutated strain also in the patients.

Answer: Mutated strains of SARS Cov-2 cannot be identified by Truenat testing or routine RTPCR test.

---

## [Editor Report · Decision Letter 1]

13 Sep 2021

Assessing a chip based rapid  RTPCR test for SARS C0V-2 detection (TrueNat assay): a diagnostic accuracy study.

PONE-D-21-10427R1

Dear Dr. Garg,

We’re pleased to inform you that your manuscript has been judged scientifically suitable for publication and will be formally accepted for publication once it meets all outstanding technical requirements.

Kind regards,

Seyed Ehtesham Hasnain

Academic Editor

PLOS ONE

Additional Editor Comments (optional):

I have gone through the revised manuscript and also the author's response to the comments of the reviewers. The manuscript was sent for revision and Authors have modified the manuscript keeping in mind the comments of the Reviewers. Conclusion section of the Abstract has been modified and Authors have updated the global data and added Indian data in the manuscript. Reference part has been updated. All grammatical and spelling errors have been taken care off.

In my view, the authors have satisfactorily addressed all the comments made by the reviewers and added all required information, and have revised the manuscript accordingly. I recommend this manuscript for publication.
---

## [Editor Report · Acceptance letter]

5 Oct 2021

PONE-D-21-10427R1 

Assessing a chip based rapid  RTPCR test for SARS C0V-2 detection (TrueNat assay): a diagnostic accuracy study. 

Dear Dr. Garg:

I'm pleased to inform you that your manuscript has been deemed suitable for publication in PLOS ONE. Congratulations! Your manuscript is now with our production department. 

Kind regards, 

on behalf of

Prof. Seyed Ehtesham Hasnain 

Academic Editor

PLOS ONE